

# MIROC6 Large Ensemble (MIROC6-LE): experimental design and initial analyses

Hideo Shiogama[1], Hiroaki Tatebe[2], Michiya Hayashi[1], Manabu Abe[2], Miki Arai[3], Hiroshi Koyama[2],
Yukiko Imada[3], Yu Kosaka[4], Tomoo Ogura[1] and Masahiro Watanabe[3]

[1]National Institute for Environmental Studies, Tsukuba, 305-8506, Japan
[2]Japan Agency for Marine-Earth Science and Technology, Yokohama, 236-001, Japan
[3]Atmosphere and Ocean Research Institute, the University of Tokyo, Kashiwa, 277-8564, Japan
[4]Research Center for Advanced Science and Technology, the University of Tokyo, Tokyo, 153-8904, Japan

*Correspondence to*: Hideo Shiogama (shiogama.hideo@nies.go.jp)

**Abstract.** Single model initial-condition large ensembles (LEs) are a useful approach to understand roles of forced responses and internal variability in historical and future climate change. Here, we produce one of the largest ensembles thus far using the MIROC6 coupled atmosphere-ocean global climate model (MIROC6-LE). The total experimental period of MIROC6-LE is longer than 76000 years. MIROC6-LE consists of a long preindustrial control run, 50-member historical simulations, 8

single forcing historical experiments with 10 or 50 members, 5 future scenario experiments with 50 members and 3 single forcing future experiments with 50 members. Here, we describe the experimental design. The output data of most of the experiments are freely available to the public. This dataset would be useful to a wide range of research communities.

  We also demonstrate some examples of initial analyses. Specifically, we confirm that the linear additivity of the forcing-response relationship holds for the 1850-2020 trends of the annual mean values and extreme indices of surface air temperature

and precipitation by analyzing historical fully forced runs and the sum of single forced historical runs. To isolate historical anthropogenic signals of annual mean and extreme temperature for 2000-2020 relative to 1850-1900, ensemble sizes of 4 and 15, respectively, are sufficient in most of the world. Historical anthropogenic signals of annual mean and extreme precipitation are significant with the 50-member ensembles in 76% and 69% of the world, respectively. Fourteen members are sufficient to examine differences in changes in annual mean values and extreme indices of temperature and precipitation between the shared

socioeconomic pathways (ssp), ssp585 and ssp126, in most of the world. Ensembles larger than 50 members are desirable for investigations of differences in annual mean and extreme precipitation changes between ssp126 and ssp119.

  Historical and future changes in internal variability, represented by departures from the ensemble mean, are analyzed with a focus on the El Niño/Southern Oscillation (ENSO) and global annual mean temperature and precipitation. An ensemble size of 31 is large enough to detect ENSO intensification from preindustrial conditions to 1951-2000, from 1951-2000 to 2051-

2100 in all future experiments, and from low- to high-emission future scenario experiments. The single forcing historical experiments with 27 members can isolate ENSO intensification due to anthropogenic greenhouse gas and aerosol forcings. Future changes in the global mean temperature variability are discernible with 23 members under all future experiments, while 50 members are not sufficient for detecting changes in the global mean precipitation variability in ssp119 and ssp126. We also confirm that these temperature and precipitation variabilities are not precisely analyzed when detrended anomalies from the

long-term averages are used due to interannual climate responses to the historical natural forcing, which highlights the importance of large ensembles for assessing internal variability.



## 1 Introduction

Internal variability in the climate system is one of the major sources of uncertainty in future climate change projections, especially at the near-term and regional scales (Hawkins and Sutton 2011, Lehner et al. 2020). Because single model initial-condition large ensembles (LEs) can provide climate scientists with useful tools for quantifying and separating the internal variability of the climate system from the responses to external forcing, some modeling centers around the world have recently produced LEs using coupled atmosphere-ocean global climate models (CGCMs). For example, the Community Earth System

Model Large Ensemble Project generated a 40-member ensemble simulation of the historical period (1920-2005) and a representative concentration pathway (RCP) 8.5 scenario (2006-2100) using the CESM1 model (CESM1-LE, Kay et al., 2015). By using CESM2, Rodgers et al. (2021) ran a 100-member ensemble for the 1850–2100 period with historical (1850–2014) and SSP3-7.0 (2015–2100) forcings (CESM2-LE). The Max Planck Institute Grand Ensemble (MPI-GE) consists of an 800-year preindustrial condition run, 100-member ensembles over the historical period (1850-2005), 3 RCPs (RCP2.6, 4.5, 8.5 for

2006-2099) and simulations with a 1%/year $CO_2$ increase (150 years) using the MPI-ESM1.1 model (Maher et al. 2019). The Swedish Meteorological and Hydrological Institute used the EC-Earth3 model to generate 50-member ensembles over the historical interval (1970-2014) and 4 shared socioeconomic pathway simulations (2015-2100; SSP1-1.9, SSP3-3.4, SSP5-3.4-OS, and SSP5-8.5) (SMHI-LENS, Wyser et al. 2021). Lin et al. (2022) computed 110-member ensemble simulations for the historical period and SSP5-8.5 scenario using the FGOALS−g3 CGCM.

In addition to historical all forcing runs, single forcing historical experiments are important for detection and attribution studies of historical climate changes and for understanding roles of internal variability in observed historical climate changes (Gillett et al. 2021, Watanabe et al. 2014, 2021, Shiogama et al. 2016). The CESM1 Single Forcing Large Ensemble Project produced 3 "all but one" type ensembles, which kept anthropogenic aerosols (20 members), biomass burning aerosols (15) and greenhouse gases (20) fixed at the 1920 condition while all other external anthropogenic and natural forcing factors evolved

following historical and RCP8.5 scenarios (Deser et al. 2020). To contribute to the Detection and Attribution Model Intercomparison Project (DAMIP, Gillett et al. 2016, 2021), which is one of the endorsed model intercomparison projects (MIPs) of Coupled Model Intercomparison Project Phase 6 (CMIP6, Eyring et al. 2016), the CNRM-CM6-1, CanESM5, GISS-E2-1-G, IPSL-CM6A-LR, MIROC6 models have generated ensembles of single forcing historical experiments considering changes in the well-mixed greenhouse gas only, the anthropogenic aerosol emissions only and the natural forcing (solar and

volcanic activities) only (these are Tier 1 experiments from DAMIP) with ensemble sizes ≥ 10. The other models provided the outputs of smaller ensembles (note that the minimum ensemble size required by DAMIP is 3). Under the auspices of the Lighthouse Activity on Explaining and Predicting Earth System Change (LHA-EPESC) initiative from the World Climate Research Programme, Smith et al. (2022) recently proposed the Large Ensemble Single Forcing Model Intercomparison Project (LESFMIP). LESFMIP calls on modeling centers around the world to enlarge the ensemble sizes of historical single

forcing experiments from DAMIP and to perform several "all but one" type experiments to improve the understanding of the causes of past climate changes on multiannual to decadal time scales.

By using the MIROC6 CGCM (Tatebe et al. 2019), we produced a large ensemble named MIROC6-LE. MIROC6-LE consists of an 800-year preindustrial condition run, 50-member historical simulations, 8 single forcing historical simulations with 10 or 50 members, 5 SSP simulations with 50 members and 3 single forcing SSP2-4.5 simulations with 50 members. This

is one of the largest LEs currently available. The aim of this paper is to describe the design of MIROC6-LE and to show some examples of analyses.



## 2 Experimental designs

We used the MIROC6 CGCM (Tatebe et al. 2019), which contributed to CMIP6. The atmospheric component has an
approximate horizontal resolution of 1.4° and consists of 81 vertical levels. The ocean component has an approximate
horizontal resolution of 1° and 63 vertical levels. We computed 50-member historical simulations (1850-2014) using the
CMIP6 forcing dataset (Table 1). The initial conditions for the atmosphere, land and ocean are taken from the different years
of the 800-year preindustial control run, which was run under the 1850 external forcing conditions (piControl) (Tatebe et al.
2019). To understand historical climate change, we performed large ensemble simulations of the DAMIP single forcing
historical experiments (1850-2020, Gillett et al. 2016). These consist of the 50-member historical simulation experiments that
used well-mixed greenhouse gases only (hist-GHG), natural forcing (solar and volcanic) only (hist-nat), and anthropogenic
aerosol only (hist-aer) , which are Tier 1 in DAMIP; and the 10 member ensembles of volcanic only (hist-volc), solar only
(hist-sol), stratospheric ozone only (hist-stratO3) and stratospheric and tropospheric ozone only (hist-totalO3) experiments,
which are Tiers 2 or 3 in DAMIP. We computed historical land-use-land-cover change only simulations (hist-lu, 10 members)
proposed by LESFMIP.

We have performed future projections (2015-2100) with 50 ensemble members under each of the 5 SSPs (Table 2): ssp585,
ssp370, ssp245 and ssp126 are the Tier 1 experiments of the Scenario Model Intercomparison Project (ScenarioMIP, O'Neill
et al. 2016), while ssp119 is Tier 2. Well-mixed greenhouse gases only (ssp245-GHG), aerosol only (ssp245-aer) and natural
only (ssp245-nat) runs under the SSP2-4.5 scenario (proposed by DAMIP) have been expanded to 50 members.

Here, we analyze the annual mean surface air temperature (T), annual mean precipitation (P), annual maximum daily
maximum surface air temperature (Tx) and annual maximum daily precipitation (Px). The monthly mean sea surface
temperature (SST) is also analyzed in Section 3.3.

## 3 Results

### 3.1 Historical experiments

Figure 1 shows the global mean changes in T and Tx for the historical simulations. Increases in greenhouse gas concentrations
(hist-GHG) lead to larger degrees of warming than from the historical runs. Cooling due to anthropogenic aerosol emissions
(hist-aer: sulfate, black carbon and organic carbon aerosols are considered) partly compensates for GHG-induced warming.
Although large volcanic activities can cause significant cooling within a few years, natural forcing does not induce long-term
trends (hist-nat, hist-sol and hist-volc). Changes in stratospheric and tropospheric ozone (hist-stratO3 and hist-totalO3) and
land-use-land-cover (hist-lu) have little effect on the global mean T and Tx.

Figure 2 presents the global mean changes in P and Px for the historical simulations. Because P is sensitive to aerosols
(Shiogama et al. 2010a,b), decreases in P due to aerosols (hist-aer of Fig. 2a) mostly compensate for the GHG-induced
increases in P (hist-GHG) (Wu et al. 2013, Shiogama et al. 2022). Therefore, the P of the historical runs has a slight trend. In
contrast, the magnitude of the decrease in Px due to anthropogenic aerosols (hist-aer of Fig. 2d) is less than the increases in Px
due to greenhouse gases (hist-GHG), which results in a positive trend of Px in the historical runs. Large volcanic eruptions
cause significant decreases in P and Px within a few years (Iles and Hegerl 2014), but volcanic and solar forcing do not cause
long-term trends (hist-nat, hist-sol and hist-volc). Changes in P and Px due to stratospheric and tropospheric ozone are small
(hist-stratO3 and hist-totalO3). It is interesting that changes in land-use-land-cover cause slightly negative trends of P (but not
Px) (hist-lu). Global mean annual mean precipitation must be equal to global mean annual mean evaporation at the surface.
Deforestation reduces evaporation and thereby precipitation (Devaraju et al. 2015). Changes in extreme precipitation are not
necessarily controlled by such a balance between precipitation and evaporation (Sugiyama et al. 2010).



Detection and attribution studies have explicitly or implicitly assumed that individual climate responses to individual forcing agents can be linearly added to obtain the total climate response to the combined forcing agents (Shiogama et al. 2013). We test this assumption of the linear additivity of the forcing-response relationship (Fig. 3). We examine the 1850-2020 trends of

T, P, Tx and Px averaged over the world and the 26 land regions defined by the IPCC (2012). Orange boxes indicate the min-max ranges of the 1850-2020 trends from the 50-member historical runs (2015-2020 are under SSP2-4.5). We randomly select one ensemble member from each of the hist-GHG, hist-aer, hist-totalO3, hist-lu, hist-sol and hist-volc experiments 1000 times, and then calculate the sum of their trends. Blue bars show the min-max ranges and the median values of the 1000 sum of the randomly sampled single forcing runs (hist-GHG + hist-aer + hist-totalO3 + hist-lu + hist-sol + hist-volc). The median values

of the sum of single forcing runs fall within the ranges of historical runs for all the regions and the global mean for all four variables, suggesting that linear additivity holds for all of these cases, at least within MIROC6-LE.

Figure 4 shows differences in regional changes in Tx (2000-2020 minus 1850-1900) and Px (percent changes from 1850-1900 to 2000-2020) between historical and hist-nat runs: these differences indicate anthropogenic influences on Tx and Px changes. When we use only runs 1-3, anthropogenic signals of Tx are statistically significant at ±5% levels of the $t$ test over

73% of the world but are not significant, for example, over the United States of America, eastern Europe and China. As the ensemble size (N) increases to N = 10 and 50, the fraction of the area where the anthropogenic TX signals are significant rises to 90% and 96%, respectively. When we use only runs 1-3 or 1-10, the fraction of the area where anthropogenic Px signals are significant are only 19% and 35%, respectively. Therefore, the ensemble sizes of N = 3 (the minimum size requested by DAMIP) or 10 (the minimum size requested by LESFMIP) are insufficient for isolating the historical anthropogenic changes

in Px over most of the world at the grid scale (150 km) using the single model ensemble. It should be noted that spatial aggregation or multimodel averages possibly improve signal-to-noise ratios.

Figure 5 shows the fractions of the world area (%) with significant differences (at ±5% levels of the $t$ test) between historical and hist-nat (i.e., anthropogenic signals) (F(historical, hist-nat)) as a function of ensemble size. As the ensemble size increases, F(historical, hist-nat) becomes larger. Area factions are larger for temperature than for precipitation and greater for mean

changes than for extremes. F(historical, hist-nat) are nearly saturated for N = 4 and 15 for T and Tx, respectively. In contrast, F(historical, hist-nat) is not saturated and rapidly increases with larger ensemble sizes for P and Px. F(historical, hist-nat) of P and Px reach 76% and 69% at N = 50, respectively. Therefore, an ensemble size of 50 is insufficient for isolating anthropogenic signals of P and Px from the natural variability over 24% and 31% of the world's area (mainly in the subtropical ocean (Fig. 4f)). In the rest of the world, anthropogenic signals of P and Px are significant with the 50 member ensembles.


### 3.2 Future experiments

Single forcing experiments under the SSP2-4.5 scenario enable us to separate future climate responses to GHG, anthropogenic aerosols and natural forcing factors (Fig. 6). Volcanic forcing is increased from the value at the end of the historical simulation period (2015) over 10 years to the same constant value prescribed for the preindustrial control simulations and then is kept

fixed (O'Neill et al. 2016). Although the future total solar irradiance is assumed to have a small negative long-term trend (Matthes et al. 2017), the effects on the long-term trends of T, Tx, P and Px are minimal. Positive responses of T, Tx, P and Px to the GHG forcing are partly compensated by the negative responses to the aerosol forcing. Because aerosol emissions gradually decrease under SSP2-4.5 (Rao 2017, Lund 2019), the negative responses of T, Tx and Px nearly disappear by 2100. In contrast, the small but nonnegligible negative responses of P to aerosol forcing remain until 2100 due to the large sensitivity

of P to aerosol forcing (Shiogama et al. 2010a, b).

Figure 7 shows changes in global mean T, Tx, P and Px under the 5 SSP scenarios. For T, Tx and Px, the differences between the scenarios are small until 2040, and after that, ssp runs with larger radiative forcing (ssp585 > ssp370 > ssp245 > ssp126





>ssp119) have greater positive anomalies. Changes in P are different from the other variables. At the end of the 21st century, the positive P anomalies from ssp370 are similar to those from ssp245, although the T anomalies from ssp370 are larger than those from ssp245. Aerosol emissions do not decrease under the SSP3-7.0 scenario but decline in the other SSP scenarios (Rao 2017, Lund 2019). It is likely that greater negative responses of P to the larger aerosol emissions compensate substantially for the positive responses of P to the GHG forcing in ssp370. Differences in aerosol emissions between SSPs (O'Neil 2014, Rao et al. 2017, Lund 2019) also induce important differences in P anomalies until the middle of the 21st century. It seems that rapid decreases in aerosol emissions due to aggressive reductions in fossil fuel consumption cause larger increases in P in ssp119 and ssp126 than in ssp585 (the fossil-fueled development scenario). The larger aerosol emissions in ssp370 (due to weak pollution control policies) would cause smaller increases in P in ssp370 than in the other ssp runs. Single forcing runs under all SSP scenarios (especially SSP3-7.0) would be useful for further analyses, but this remains the focus of future work.

The top panels of Fig. 8 show differences in changes in Px (percent changes from 1850-1900 to 2050-2100) between ssp585 and ssp126. When we use only runs 1-3, differences in Px are statistically significant over 49% of the world's area but are not, for example, significant over the United States of America and Europe. As the ensemble size increases to 10 and 50, the fractions of areas with significant differences rise to 77% and 90%, respectively. When we analyze the differences in changes in Px between ssp126 and ssp119 (the bottom panels of Fig. 8), only 38% of the globe shows significant differences even when we have 50 member ensembles. When the ensemble size is limited to 10 or 3, the areas where differences are significant cover only 11% and 7%, respectively.

Figure 9 shows the fractions of areas with significant differences in T, Tx, P and Px changes (changes from 1850-1900 to 2050-2100) between the SSP experiments as a function of ensemble sizes (F(sspXXX, sspYYY)). For T, F(ssp126, ssp119), F(ssp245, ssp126), F(ssp370, ssp126) and F(ssp585, ssp126) exceeded 80%, even when only 3 members were available. For Tx, when F(ssp126, ssp119), F(ssp245, ssp126), F(ssp370, ssp126) and F(ssp585, ssp126) have 10 ensemble members, the areas are 84%, 94%, 97% and 99%, respectively. Therefore, 10 members are sufficient to examine differences in T and Tx changes over most of the world. Fourteen and 13 members are necessary for F(ssp585, ssp126) of P and Px to exceed 80%, respectively. With N = 25, the F(ssp370, ssp126) of P and Px reach 80%. For the F(ssp245, ssp126) of P and Px, N ≥ 38 and 46 are necessary to exceed 70%. The F(ssp126, ssp119) of P and Px are less than 50% for N = 50. Therefore, larger ensembles are desirable to investigate the differences in mean and extreme precipitation changes between ssp126 and ssp119, which were designed as experiments relevant to the 2 °C and 1.5 °C goals of the Paris Agreement (O'Neill et al. 2016).

### 3.3 Changes in internal variability

The internal variabilities in the historical (1951-2000) and future (2051-2100) experiments of MIROC6-LE are examined. Climate variability consists of externally forced change and internal variability, and isolating internal variability under transient forced changes with low errors requires large ensembles (Maher et al. 2018, Milinski et al. 2020, Lee et al. 2021) or spatiotemporal analysis methods (e.g., Wills et al. 2020). In this section, two conventional methods are compared for determining the internal variability component of an area-averaged variable: (1) a single-member estimate using linearly detrended anomalies relative to the seasonal climatology over 50 years and (2) a multimember estimate using departures from the ensemble mean changing with time in response to external forcing. Hereafter, we represent the internal variability components of the first and second methods with "a" and "i": for instance, Xa and Xi for a variable X, respectively. While the first method is often used to analyze observational datasets and climate model outputs (e.g., Kim et al. 2014, Capotondi et al. 2020), the second method, which requires a large ensemble, is more appropriate for determining the internal variability component since it is not contaminated by residuals from detrending methods. Based on the multimember estimate with 50 members, we term 50 member ensemble averages "the best estimate" in this study. As the reference state for the forced



responses, the 800-year piControl is also analyzed, where Xi is the departure from the seasonal climatology over 800 years as the external forcing is constant for the year 1850, while the 50 members of Xa are derived from randomly selected 50-year segments. To focus on the interannual timescale, a 10-year high-pass Butterworth filter is applied.

The amplitude of the El Niño/Southern Oscillation (ENSO; Timmermann et al. 2018), the dominant interannual SST variability in the tropical Pacific Ocean, is examined based on the standard deviations of SSTa and SSTi averaged over the Niño-3.4 region (170°–120°W, 5°S–5°N; Trenberth 1997) in boreal winter (December-January-February). It has been reported that MIROC6 is one of the CMIP6 climate models that better simulates various key ENSO properties, such as spatial structure, global teleconnections, nonlinearity and ENSO dynamics (Tatebe et al. 2019, Hayashi et al. 2020, Fasullo 2020, Planton et al. 2021). In each MIROC6-LE experiment (Figure 10a), the Niño-3.4 SSTa standard deviation varies among the ensemble members depending on the initial condition, but its 50 member ensemble average (shown with an open circle) approximates the best estimate (shown with a cross mark), implying that the forced response of the Niño-3.4 SST is not sensitive to interannual external forcing, such as volcanoes. Thus, the single-ensemble estimate can decompose the internal ENSO variability properly. Figure 10a also shows that the ENSO amplitude increases from the piControl to the historical (1951-2000) experiments and is further enhanced in all the future projections (ssp119, ssp126, ssp245, ssp370, ssp585) for 2051-2100. The increased amplitude from the preindustrial condition of 1850 to 1951-2000 is consistent with the past ENSO changes inferred by McGregor et al. (2013). The future strengthening of ENSO SST variability agrees with the majority of CMIP6 climate models under transient warming scenarios (Fredriksen et al. 2020, Cai et al. 2022), while equilibrated warmer climates may weaken ENSO (Challahan et al. 2021). The DAMIP experiments indicate that the increased GHG forcing contributes to ENSO amplification in the historical and ssp245 experiments, while the aerosol forcing also enhances ENSO in the historical experiment in MIROC6, as noted by Maher et al. (2022, in press).

How many ensemble members are needed for approximating the best estimate? In Figure 10a, the 50-member ensemble averages (opened circles) correspond well to the best estimates (cross marks), but such a large ensemble requires a high computing resource and thus is not available for many climate models. To provide an efficient way to analyze internal variabilities in practice, the accuracy of ensemble averages with smaller ensemble sizes (e.g., 10 members) is evaluated. Here, the robustness of the ENSO amplitude differences among the MIROC6-LE experiments is tested with respect to the ensemble size (N). The uncertainty in the single-member estimate is shown in Figure 10b-d, where the 95% ranges between the 2.5th and 97.5th percentiles are calculated from 1000 pseudoensembles, which are produced for each N from 1 to 50 by resampling the 50 Niño-3.4 SSTa standard deviations randomly with replacement (Lee et al. 2021). The uncertainty range becomes larger with smaller values for N, and then two ranges in different experiments may overlap at a specific small N, where the difference between two ensemble averages is possibly due to a lack of ensemble members. The minimum sizes of N needed for separating the two experiments beyond their 95% ranges are summarized in Supplementary Data S1. To separate the historical from piControl experiments, N = 3 is at least required. The amplitude differences from historical to ssp370 and to ssp245 and ssp585 are detectable with N≥3 and N≥4, respectively, while much larger ensembles are required for ssp126 (N≥20) and ssp119 (N≥33). An ensemble with N≥46 can distinguish ssp370 from ssp245. However, the differences between ssp119 and ssp126 and among ssp245, ssp370, and ssp585 are not clearly obtained by N=50, thus a larger ensemble is required. In the DAMIP experiments, strengthening of ENSO from piControl can be detected with N≥2 for ssp245-GHG, N≥7 for hist-GHG, and N≥23 for hist-aer. The ENSO amplitude increase is also robustly detected from hist-GHG to the historical runs with N≥22 and from ssp245-GHG to ssp245 with N≥37, implying that external forcings other than GHG (e.g., aerosols) and the nonlinear relationship of forced responses may contribute to strengthening ENSO.

The ENSO amplitude uncertainty in the multimember estimate with N is also shown in Figure 10e-g, where the Niño-3.4 SSTi is defined as the departure from the ensemble mean of N members (3≤N≤50) randomly resampled with replacement. In general, a small N (e.g., N=3) results in underestimating the internal variability amplitude, as the ensemble mean includes a higher residual variability (Milinski et al. 2020). For the ENSO amplitude, the results are basically the same as the single-





member estimate case (Figures 10b-d), except for the detailed numbers. For example, N≥5 is required for separating historical runs from the piControl run, and N=8 is large enough to detect the differences from historical runs to ssp245 (N≥7), ssp370 (N≥6), and ssp585 (N≥8). A larger N is required to identify differences among historical runs and ssp126 (N≥19) and ssp119 (N≥31). The ENSO strengthening from piControl to hist-GHG and hist-aer are detectable with N≥8 and N≥27, respectively. Importantly, an ensemble with at least N=30 is necessary to avoid degrading the internal variability amplitude derived by the multimember estimate.

The internal variabilities in the global annual mean surface air temperature (Ta and Ti) and precipitation (Pa and Pi) shown in Figures 6 and 7 are analyzed in the same manner as the ENSO amplitude. Figures 11a and 12a provide the amplitude dependence on the experiments. The variability is lower in the piControl experiment and higher in the high emission scenarios (ssp245, ssp370, ssp585). It also increases from piControl to hist-GHG and ssp245-GHG. Note that the single-member estimates (Figures 11b-d and 12b-d) overestimate the global mean T and P variabilities than the best estimates in the historical and hist-nat runs, in contrast to the ENSO amplitude. This is because the externally forced change at the interannual and decadal timescales driven by natural forcing, such as volcanic eruptions (Figures 1a and 2a), cannot be separated from the internal variability and thus contaminates the Ta and Pa variabilities. Therefore, the multimember estimate (Ti and Pi) is necessary for distinguishing the internal variability components from the forced responses in the global mean T and P of the historical and hist-nat experiments (Figures 11e-g and 12e-g).

Based on the multimember estimate, the minimum ensemble size for detecting the increase in the global mean Ti standard deviation is N=6 for the piControl to historical runs, N=7 for the historical to ssp370 runs, N=8 for the historical to ssp245 and ssp585 runs, and N=23 for the historical to ssp119 and ssp126 runs. The differences between ssp119 and ssp126 and among ssp245, ssp370, and ssp585 are not significant even for N=50. Interestingly, the global mean Ti standard deviations tend to be higher in ssp370 than in ssp585, similar to ENSO, indicating the critical role of ENSO variability on the global mean air temperature variance (e.g., Thompson et al. 2009, Hu and Fedorov 2017). In contrast, the global mean Pi standard deviations overlap with each other for about 95% of their ranges so that the differences among historical, ssp119 and, ssp126 runs are not robust even for N=50. However, when N is greater than or equal to 16, the increase is significant from historical scenarios to the high-emission scenarios (ssp245, ssp370, ssp585), for N=50 from ssp126 to ssp245, and for N=47 and 33 for ssp245 to ssp370 and ssp585, respectively. Unlike the global mean Ti and ENSO variabilities, the Pi variability is higher in ssp370 and ssp585 than in ssp245. It is significant for N≥10 that the increased variability in both the global mean Ti and Pi is driven by GHG (hist-GHG and ssp245-GHG), but the amplitudes of that variability in the other DAMIP experiments remain indistinguishable in the piControl run, even with N=50. Thus, a larger ensemble and longer piControl simulation are desirable for detecting changes in the global mean T and P internal variability amplitude from the piControl to hist-aer, hist-nat, ssp245-aer, and ssp245-nat simulations.

In summary, MIROC6-LE with N=50 is useful for detecting changes in the internal variability of ENSO and global mean T and P anomalies between historical and future experiments and between high- and low-emission scenarios. The impact of aerosols on ENSO amplitude is also detectable in MIROC6. However, it is still challenging to distinguish these changes between high-emission scenarios or between low-emission scenarios, implying the importance of implementing spatiotemporal analysis methods (e.g., Wills et al. 2020) or generating larger ensembles (e.g., Maher et al. 2019, Rodgers et al. 2021, Lin et al. 2022). The mechanisms underlying the clearly detected changes in this study and the changes in other internal variabilities will be reported in future works.



## 4 Summary and discussion

Here, we have explained how MIROC6-LE was designed. We have also shown the results of the initial analyses. Specifically, we find that the linear additivity of the forcing-response relationship holds for the 1850-2020 trends of T, Tx, P and Px by analyzing the historical all forcing runs and the sum of single forcing historical runs. The ensemble sizes of 4 and 15 are

enough to isolate historical anthropogenic signals (differences between historical and hist-nat runs) of T and Tx in most of the world. Although historical anthropogenic signals of P and Px are significant with the 50 member ensembles in approximately 70% of the world, larger ensembles are necessary in other areas (mainly in the subtropics). The 50 member simulations of MIROC6-LE are sufficient to analyze differences in regional T and Tx changes between ssp126 and ssp119 (scenarios relevant to the 2 °C and 1.5 °C goals of the Paris Agreement) but not sufficient to obtain significant differences in P and Px changes

over more than half of the world. Atmospheric global climate models (AGCMs) are more cost-effective tools to produce large ensembles (e.g., 100 members) for future climate change projections at given warming levels and for event attribution studies than CGCMs (Shiogama et al. 2016, 2020, Mizuta et al. 2017, Mitchell et al. 2017, Imada et al. 2017, Stone et al. 2019, Fujita et al. 2020, Nosaka et al. 2021). Combined analyses of AGCM LEs (e.g., Mitchell et al. 2017, Shiogama et al. 2019) and CGCM LEs could be useful for discussions of differences in precipitation changes between the 2 °C and 1.5 °C warmer

climates, but it should be noted that climate change projections can be significantly different between AGCM and CGCM simulations (Uhe et al. 2021).

We also analyzed the historical and future changes in ENSO and the internal components of interannual global mean T and P variabilities using MIROC6-LE. The interannual variabilities are the 10-year high-pass filtered components of departures from the ensemble mean so that interannual and decadal responses to the natural forcing can be eliminated, and they are

compared with detrended anomalies relative to the long-term mean. The future ENSO intensification over the 50-year window of 2051-2100 is detectable with ensemble sizes of 6 for ssp370 (minimum of the 5 scenarios examined) and 31 for ssp119 (maximum), and the ensemble size of 27 can isolate the contributions of the historical GHG and aerosol forcings to the ENSO intensification during the historical period. For detecting the future amplification in the interannual T variability, the ensemble sizes of 8 and 23 are sufficient in the high-emission scenarios (ssp245, ssp370, ssp585) and in the low-emission scenarios

(ssp119, ssp126), respectively, but the differences between the low- and high-emission scenarios cannot be isolated even with 50 members. As these unforced global mean T variability changes basically follow the ENSO amplitude changes (cf. Thompson et al. 2009), a better understanding of ENSO projections is key for reducing the uncertainty of global temperature projections. The P variability amplification from the historical runs can be detected in the high-emission scenarios with 16 members, but 50 members are not sufficient in the low-emission scenarios. It is worth noting that an ensemble with at least 30

members is necessary to avoid underestimating the amplitude of internal variability relative to the ensemble mean. Furthermore, the historical natural forcing results in an overestimation of the global mean T and P variabilities for interannual timescales when the detrended anomalies are used, highlighting the importance of using the departures from the ensemble mean for discussing changes in the internal variability.

MIROC6-LE is one of the largest LEs to date. The total simulation length of MIROC6-LE is 76750 years, which is much

larger than that of other LEs, e.g., CESM2-LE (25100 years), MPI-GE (58800 years) and SMHI-LENS (19450 years), and is comparable       to       the       DAMIP       experiments       of       CanESM5       (72515       years, https://pcmdi.llnl.gov/CMIP6/ArchiveStatistics/esgf_data_holdings/DAMIP/index.html).       Because       MIROC6-LE       is       an evolving ensemble, additional experiments will be included in the future. We are publishing the output data of MIROC6-LE via the Earth system federation grid. When this paper was submitted, the output data of all the experiments except for runs 1-

10 of hist-lu and runs 11-50 of hist-aer and ssp245-aer had been published. Because the experimental designs of MIROC6-LE are consistent with those of the other MIPs, it is easy to compare the outputs of MIROC6-LE with the other models. We hope that the publicly available output data of MIROC6-LE facilitate research in a broad range of communities.



**Acknowledgments**

This work was supported by the Advanced Studies of Climate Change Projection (SENTAN, JPMXD0722680395) and Grants-in-Aid for Scientific Research (JP21H01161, JP21K13993, JP22H01302 and JP23H01241) of the Ministry of Education, Culture, Sports, Science and Technology of Japan. The MIROC6 simulations were performed using the Earth Simulator in JAMSTEC and the NEC SX in NIES.

**Code and data availability**

The output data of all the experiments except for runs 1-10 of hist-lu and runs 11-50 of hist-aer and ssp245-aer are available via the Earth system federation grid (https://esgf-node.llnl.gov/search/cmip6/):https://doi.org/10.22033/ESGF/CMIP6.894 (Shiogama 2019), https://doi.org/10.22033/ESGF/CMIP6.898 (Shiogama et al. 2019), https://doi.org/10.22033/ESGF/CMIP6.5711 (Tatebe and Watanabe 2018a),

https://doi.org/10.22033/ESGF/CMIP6.5603 (Tatebe and Watanabe 2018b). The output data of runs 1-10 of hist-lu and runs 11-50 of hist-aer and ssp245-aer will be published within one year. Before that those data can be available from the corresponding author upon reasonable request. The codes are also available from the corresponding author.

**Author contributions**

HS and HT performed the experiments. M. Abe, M. Arai and HK contributed to the experimental setup, data publication and construction of the data storage sever, respectively. HS and MH analyzed the data and wrote the first draft of the manuscript. YI, YK, TO and MW provided insights into the results. MW led the project. All authors contributed to improving the manuscript.

**Competing interests**

The contact author has declared that no authors have any competing interests.

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





**Table 1.** List of historical experiments. Note that an 800-year preindustrial control run under the 1850 forcing condition is also included in MIROC6-LE.

| Exp name | Forcing | Ens. Size | Start Year | End Year | MIP (Tier) |
|---|---|---|---|---|---|
| historical | Anthropogenic and natural external forcing | 50 | 1850 | 2014 | CMIP6 DECK (1) LESFMIP (3) |
| hist-nat | Natural external forcing only | 50 | 1850 | 2020 | DAMIP (1) LESFMIP (3) |
| hist-GHG | Well-mixed GHG only | 50 | 1850 | 2020 | DAMIP (1) LESFMIP (1) |
| hist-aer | Anthropogenic aerosols only | 50 | 1850 | 2020 | DAMIP (1) LESFMIP (1) |
| hist-volc | Volcanic-only | 10 | 1850 | 2020 | DAMIP (3) LESFMIP (1) |
| hist-sol | Solar-only | 10 | 1850 | 2020 | DAMIP (3) LESFMIP (1) |
| hist-stratO3 | Stratospheric ozone only | 10 | 1850 | 2020 | DAMIP (2) |
| hist-totalO3 | Stratospheric and tropospheric ozone only | 10 | 1850 | 2020 | DAMIP (3) LESFMIP (1) |
| hist-lu | Land use land cover change only | 10 | 1850 | 2020 | LESFMIP (1) |



**Table 2.** List of future experiments


| Exp name | Forcing | Ens. Size | Start Year | End Year | MIP (Tier) |
|---|---|---|---|---|---|
| ssp585 | SSP5-8.5 | 50 | 2015 | 2100 | ScenarioMIP (1) |
| ssp370 | SSP3-7.0 | 50 | 2015 | 2100 | ScenarioMIP (1) |
| ssp245 | SSP2-4.5 | 50 | 2015 | 2100 | ScenarioMIP (1) |
| ssp126 | SSP1-2.6 | 50 | 2015 | 2100 | ScenarioMIP (1) |
| ssp119 | SSP1-1.9 | 50 | 2015 | 2100 | ScenarioMIP (2) |
| ssp245-nat | Natural only of SSP2-4.5 | 50 | 2021 | 2100 | DAMIP (3) |
| ssp245-GHG | GHG only of SSP2-4.5 | 50 | 2021 | 2100 | DAMIP (2) |
| ssp245-aer | Anthropogenic aerosols only of SSP2-4.5 | 50 | 2021 | 2100 | DAMIP (3) |



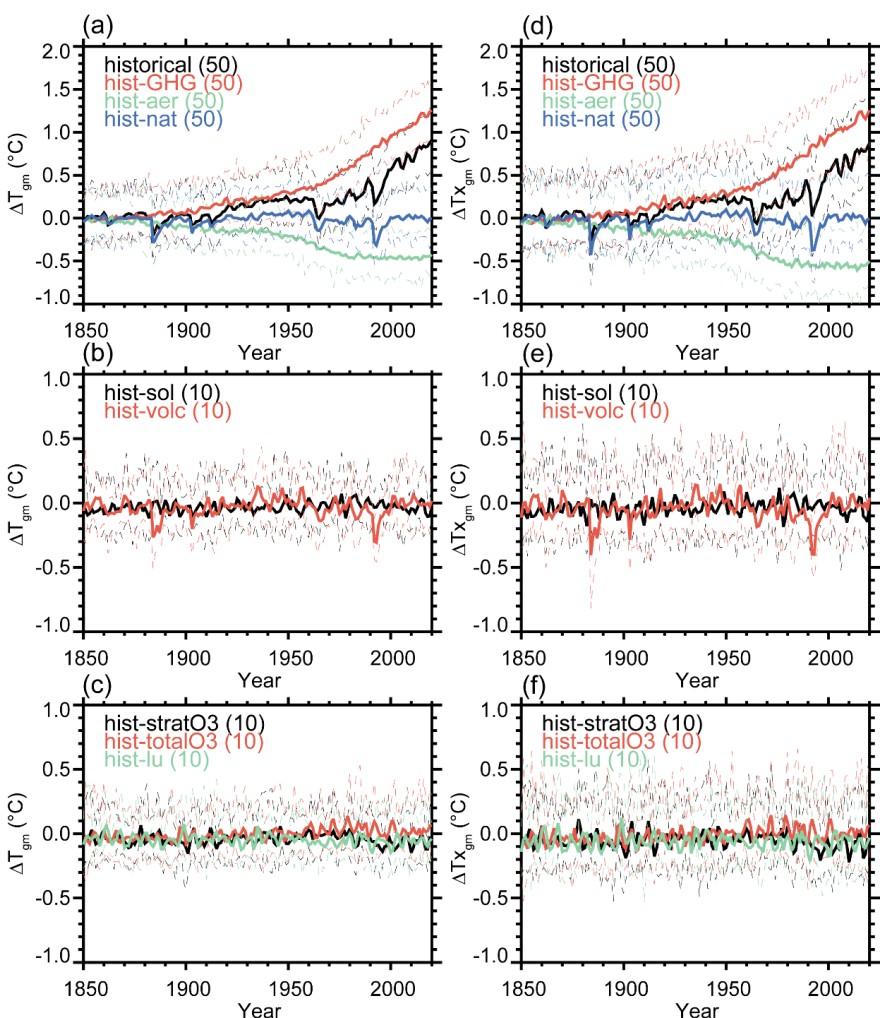

**Figure 1: Changes in global mean T (left, °C) and Tx (right, °C) of historical all and single forcing experiments relative to the 1850-1900 averages of the ensemble-mean historical all forcing simulations. Solid lines are the ensemble means. Thin dashed lines denote the minimum and maximum values of the ensemble members. Numbers in parentheses indicate the ensemble sizes.**


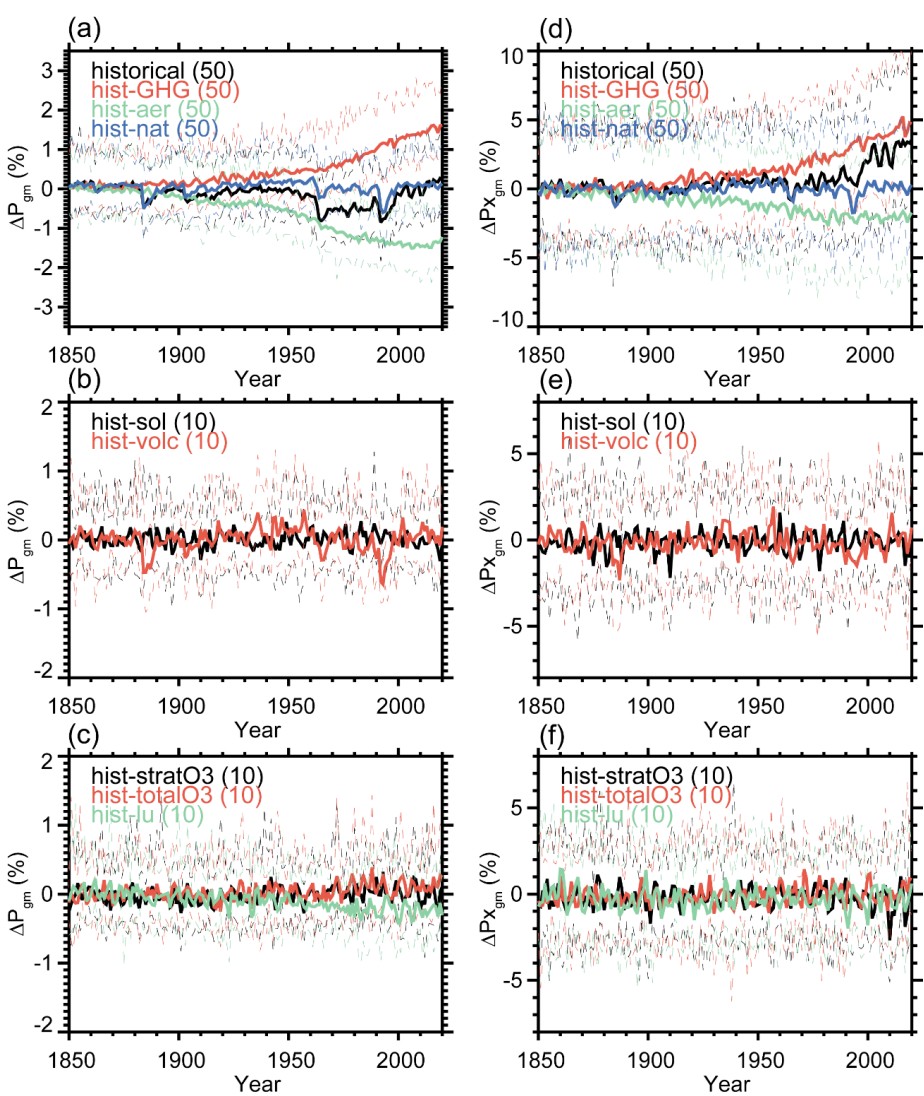

**Figure 2: Global mean changes in P (left, %) and Px (right, %) of historical all and single forcing experiments relative to the 1850-550 1900 averages of the ensemble-mean historical all forcing simulations. Solid lines are the ensemble means. Thin dashed lines denote the minimum and maximum values of the ensemble members. Numbers in parentheses indicate the ensemble sizes.**



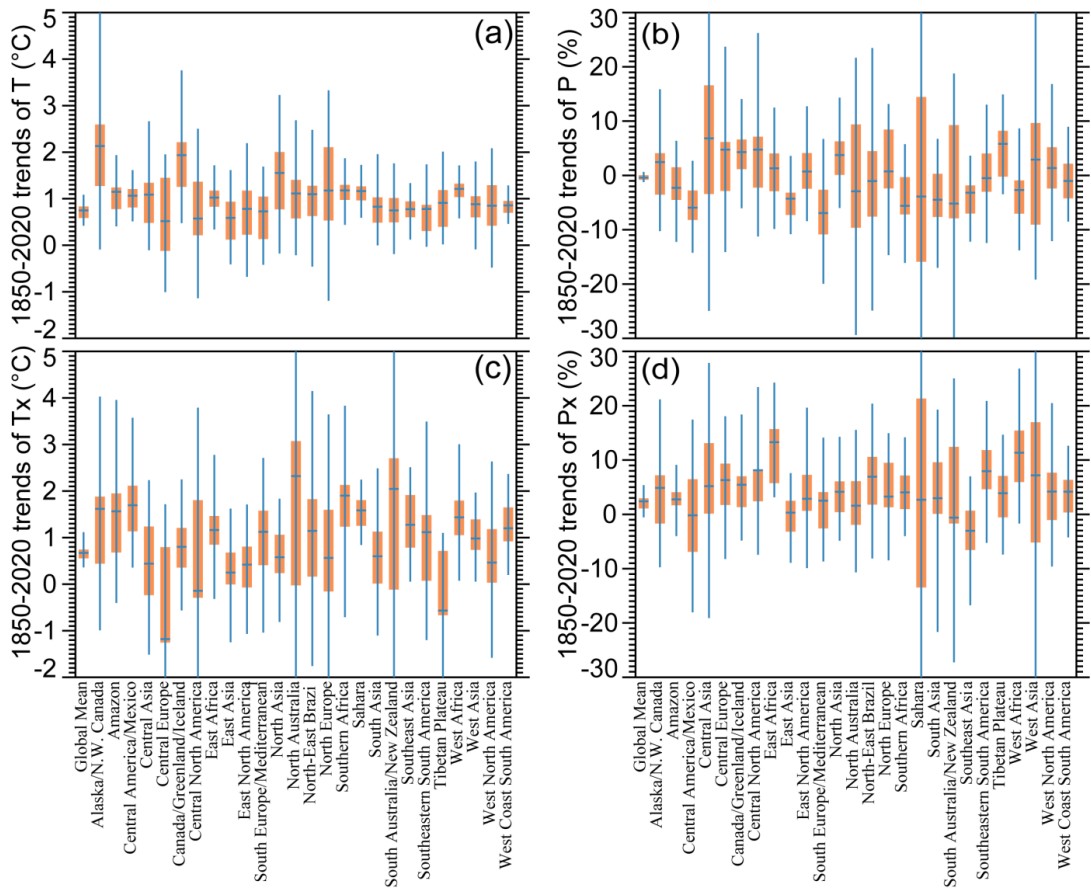

**Figure 3: Additivity tests of historical experiments for (a) T (°C), (b) P (%), (c) Tx (°C) and (d) Px (%). Orange boxes indicate the min-max ranges of the 1850-2020 trends of the 50 member historical runs. These trends are calculated for the global mean or regional averages over the 26 regions defined by the IPCC (2012). Blue bars show the min-max ranges and the median values of 1000 sum of randomly sampled single forcing runs (hist-GHG + hist-aer + hist-totalO3 + hist-lu + hist-sol + hist-volc).**



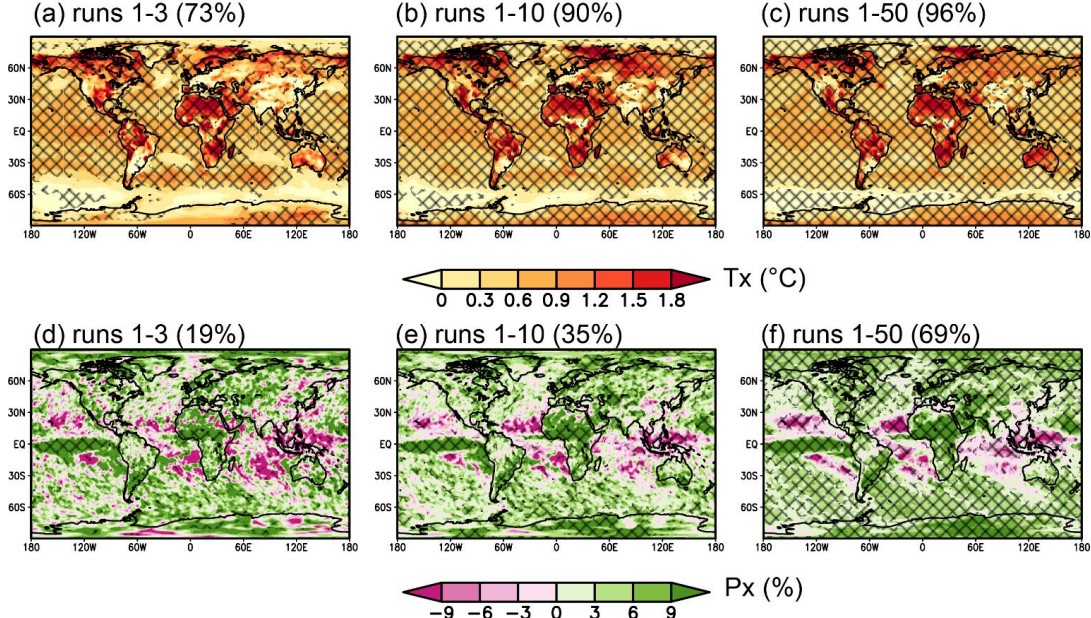

**Figure 4: (Top) Differences in changes in Tx (°C, 2000-2020 minus 1850-1900) between historical and hist-nat runs. Shading shows**
**the ensemble mean of (a) runs 1-3, (b) runs 1-10 and (c) runs 1-50. Hatching denotes the regions where the differences are significant**
**at 5% levels of the $t$ test. Parentheses indicate the fraction of the area with significant differences. (Bottom) The same as the top**
**panels but for differences in changes in Px (percent changes of 2000-2020 relative to 1850-1900) between historical and hist-nat runs.**





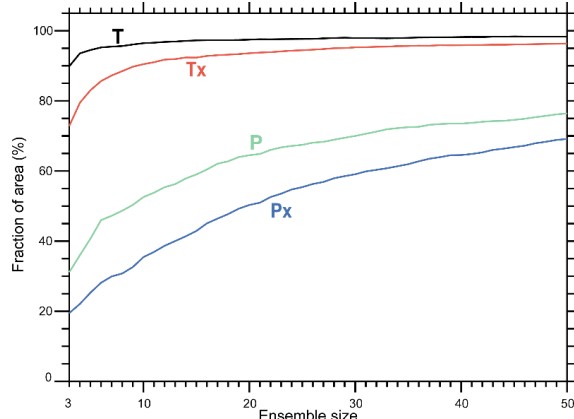

**Figure 5: Fraction of the world area (%) with significant differences (at 5% levels of the _t_ test) between historical and hist-nat as a function of ensemble size. Black, red, green and blue lines are T, Tx, P and Px, respectively.**

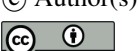

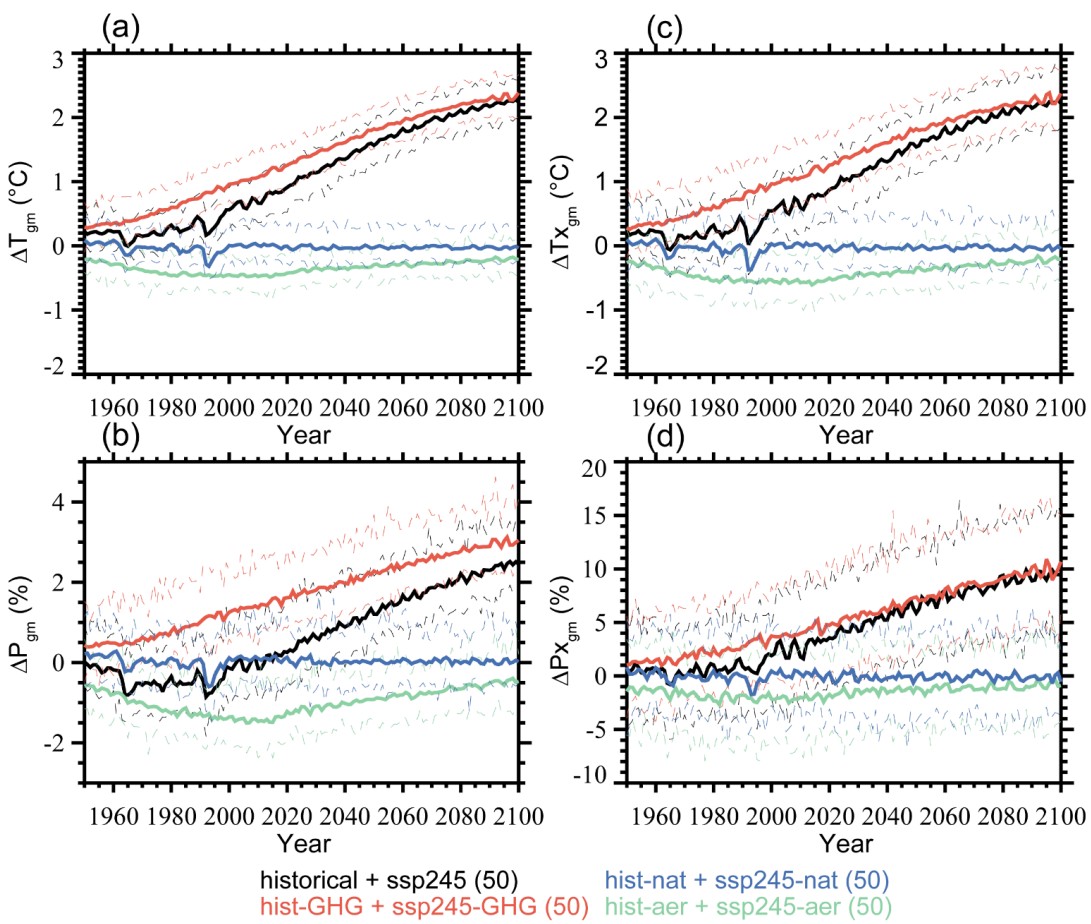

**Figure 6: Historical and future changes in global mean (a) T (°C), (b) P (%), (c) Tx (°C) and (d) Px (%) relative to the 1850-1900**
**averages due to single forcing factors. Solid lines are the ensemble averages of "historical + ssp245" (black), "hist-GHG + ssp245-GHG" (red), "hist-aer + ssp245-aer" (green) and "hist-nat + ssp245-nat" (blue). Thin dashed lines denote the minimum and maximum values of the ensemble members. Numbers in parentheses indicate the ensemble sizes.**



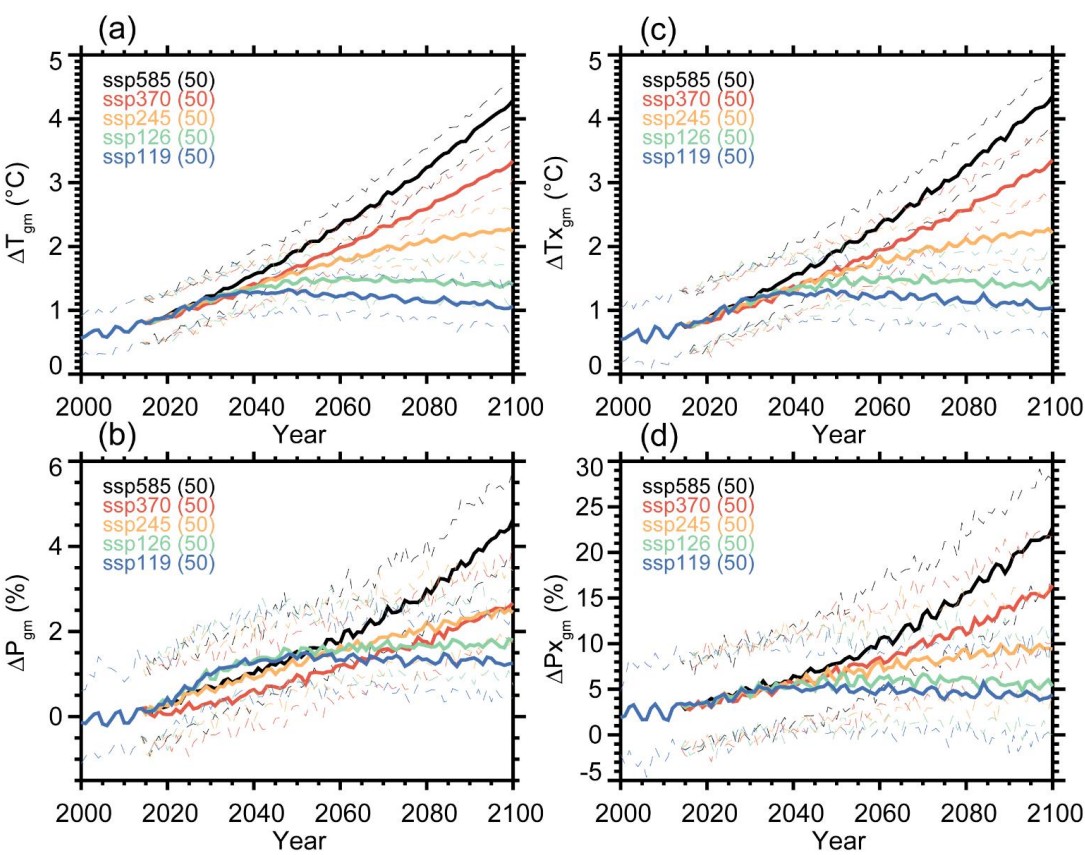

**Figure 7: Future changes in global mean (a) T (°C), (b) P (%), (c) Tx (°C) and (d) Px (%) relative to the 1850-1900 averages. Solid lines are the ensemble averages of ssp585 (black), ssp370 (red), ssp245 (orange), ssp126 (green) and ssp119 (blue). Thin dashed lines denote the minimum and maximum values of the ensemble members. Numbers in parentheses indicate the ensemble sizes.**

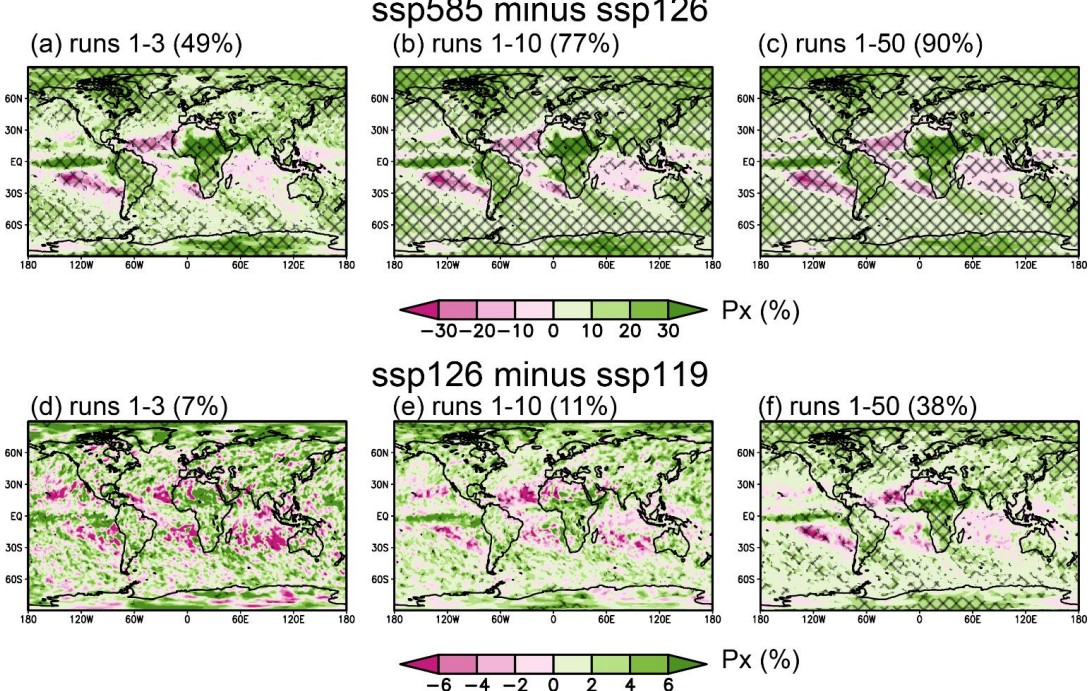

Figure 8: (Top) Differences in changes in Px (percent changes of 2050-2100 relative to 1850-1900) between ssp585 and ssp126 runs. Shading shows the ensemble mean of (a) runs 1-3, (b) runs 1-10 and (c) runs 1-50. Hatching denotes the regions where the differences are significant at 5% levels of the *t* test. Numbers in parentheses indicate the fraction of the areas with significant differences. (Bottom) The same as the top panels but for differences between ssp126 and ssp119.



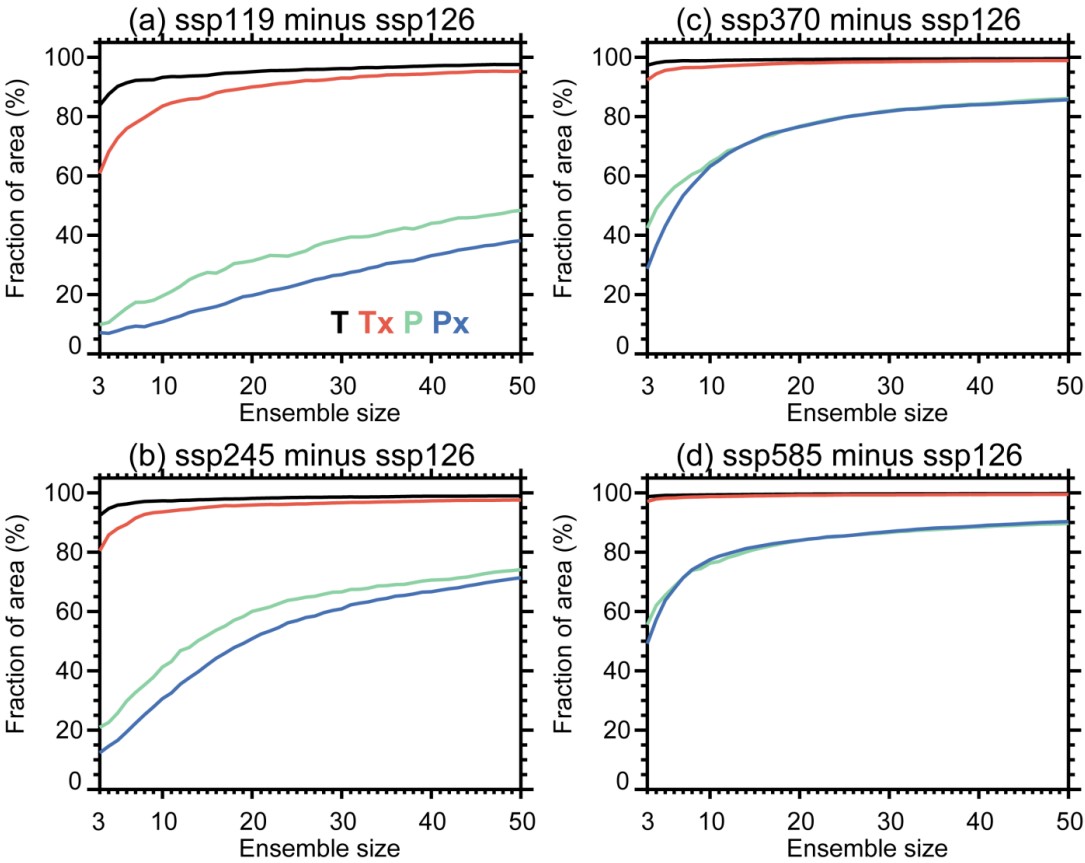

**Figure 9: Fraction of the world area (%) with significant differences (at 5% levels of the *t* test) between (a) ssp126 and ssp119, (b) ssp245 and ssp126, (c) ssp370 and ssp126, and (d) ssp585 and ssp126 as a function of ensemble size. Black, red, green and blue lines are T, Tx, P and Px, respectively.**

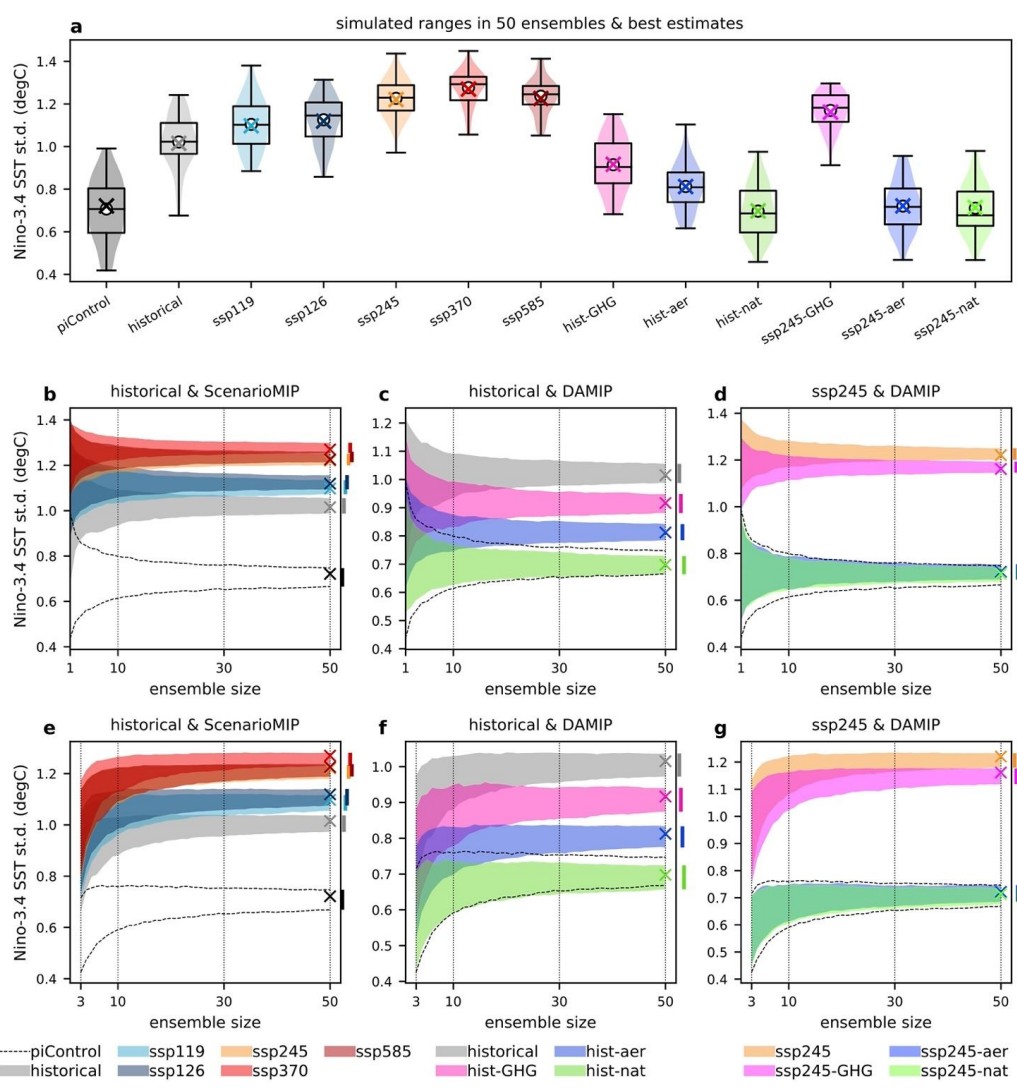


**Figure 10: The standard deviations of the 10-year high-pass filtered Niño-3.4 SSTa and SSTi averaged in December-January-February. (a) The best estimates derived from the 50 members of SSTi (cross marks) and the single-member estimates derived from each member of SSTa (box-whisker plots for the ensemble medians, interquartile ranges, and minimum-maximum ranges; opened circles for the ensemble mean; shadings for probability density functions). The ensemble-size (N) dependence of the 95% ranges of (b-d) the ensemble means of randomly selected SSTa from the 50 members and (e-g) the SSTi as the departures from the ensemble means of randomly selected members. The cross marks indicate the best estimates. The vertical bars outside of each panel show the 95% ranges at N=50.**




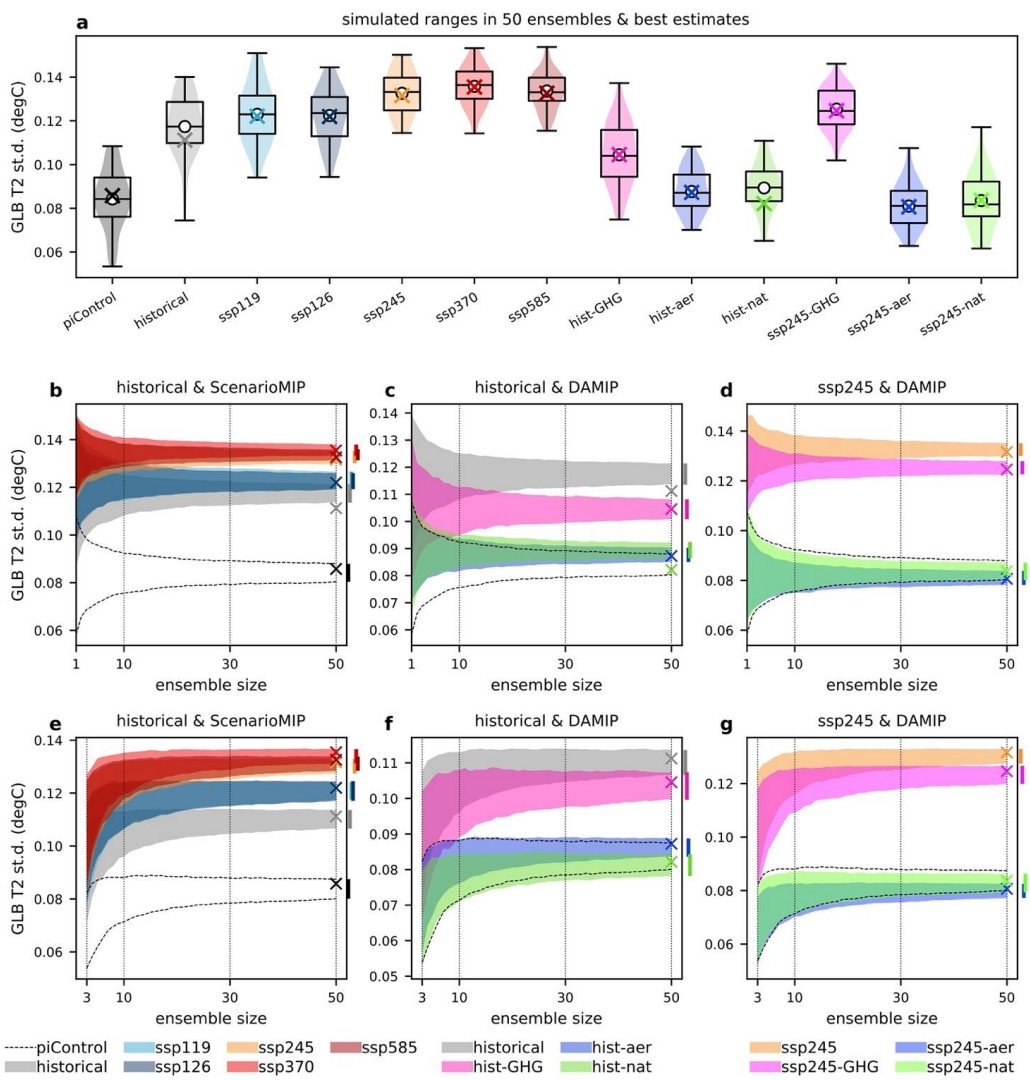

**Figure 11: As in Figure 10 but for the 10-year high-pass filtered global annual mean T (°C).**



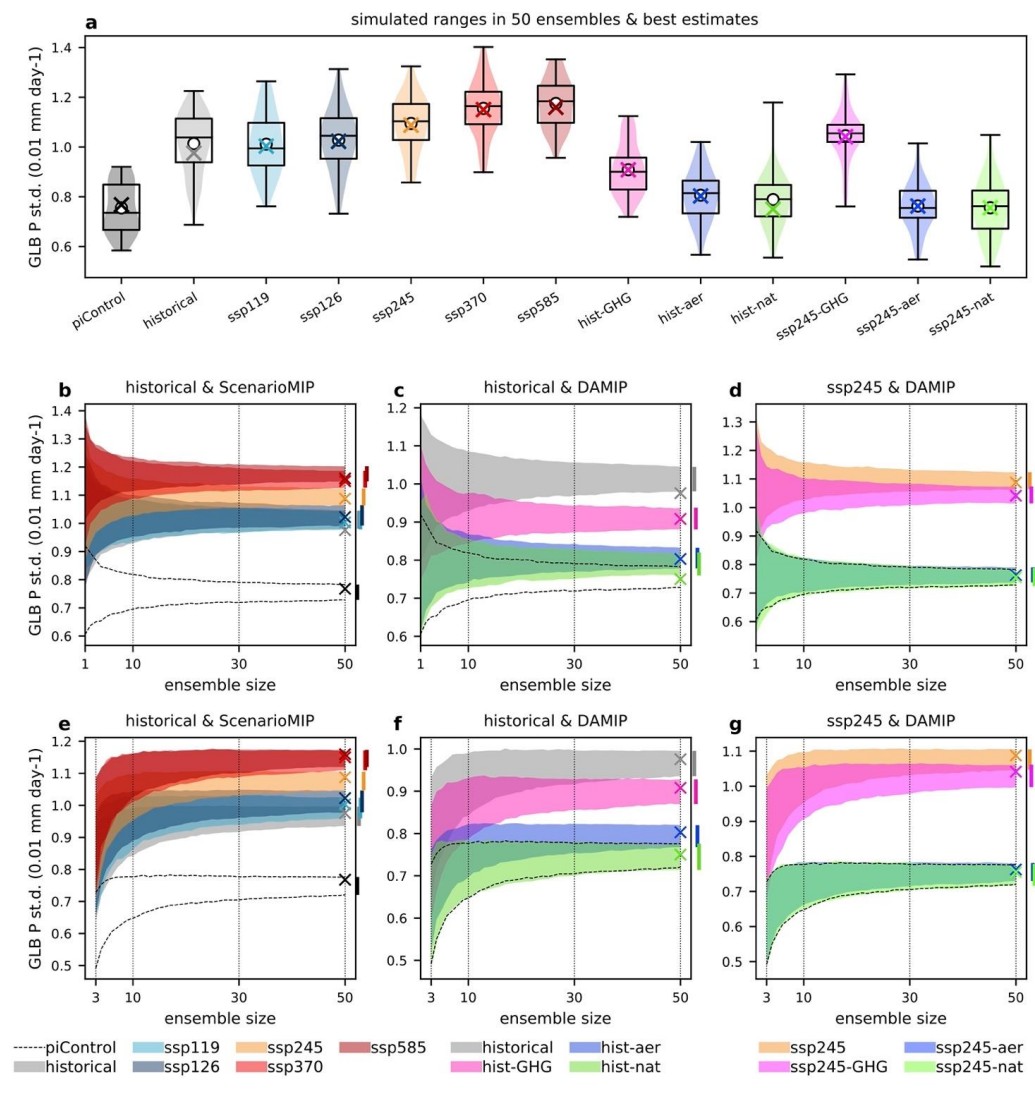


**Figure 12: As in Figure 10 but for the 10-year high-pass filtered global annual mean P (0.01 mm/day).**