# Peer review of "MIROC6 Large Ensemble (MIROC6-LE): experimental design and initial analyses"

_Earth System Dynamics, 2023_

## Author Comment (AC1)

Reply to the reviewer #1

This study introduces an impressive new dataset that is available for community use with the MIROC6 model. This dataset consists of large ensembles of historical and future projections with multiple scenarios as well as single forcing simulations. The primary purpose of the paper is to introduce these simulations but it also presents some cursory, but useful, analyses of changes in global temperature, precipitation and their extremes and some assessment of non-linearity in the single forcing simulations and analysis of the number of members required to detect changes or differences between scenarios. Overall, I think this is a useful and well written study that introduces this important new dataset and I have only minor recommendations to consider before publication.

Thank you very much for your useful comments.

l83: It could be worth providing a bit more information about the piControl and the initialization dates. Firstly, it could be worth stating whether the piControl is still drifting at this stage and some more specifics about the initialization dates e.g., were they spaced by a certain number of years?

The piControl run is stable (Fig. 3 of Tatebe et al. 2019). We will involve the list of initial conditions as a table.

L87: It can sometimes be a bit confusing where biomass burning aerosols are represented. I'm assuming that they are included in the anthropogenic aerosol contribution? Even though there is a natural component to that. It might be worth being clear about this.

Although biomass burning aerosol emissions include both anthropogenic and natural components, anomalies from piControl (involving the 1850 emission of biomass burning aerosols) can be used to estimate the anthropogenic component.

l120-126 and Fig 3: I'm not sure what the motivation is for doing this assessment of non-linearity by using only samplings of 1 member. It may be that there is a true non-linearity but

you can only see it in the ensemble means.   You could do the same analysis but sample N members with replacement from each ensemble, where N is your original ensemble size, to determine whether there are any non-linearities that can be detected with the ensemble means.

Thank you for the useful advice. For the revised version, we have computed the max-min ranges of 1000 ensemble average values of randomly sampled ensemble members with replacement. The range of the ensemble mean of historical runs overlap with the range of the sum of the ensemble average of individual forcing experiments. Therefore we have not found any statistical evidence of non-linearities in the ensemble mean values.

l137: At the introduction to Fig 5, it might help readers to remind them what time period is being considered.   I think it's 2000-2020 minus 1850-1900?

We will denote "2000-2020 minus 1850-1900" in the caption of Fig. 5.

l147-151: I think this text is describing the behavior of the hist-nat+ssp245-nat run in Figure 6, but it's not entirely clear.   Maybe reference that part of the figure when referring to the solar and volcanic contributions.

This text is describing the solar and volcanic forcing in the future simulations (2015-). We will made it clear in the text as "For future simulations from 2015, volcanic …" and will draw only the period of 2015-2100 in Fig. 6.

l210: It seems like another possibility beyond the interannual external forcings is inacuracies in the use of a linear trend?   If so, that could be mentioned too.

Thank you for the question. First, we would like to clarify the sentence, "… the forced response of the Niño-3.4 SST is not sensitive to interannual external forcing, such as volcanoes". As seen in Figures 1 and 2, the ensemble averages (forced responses) of the global mean annual temperature and precipitation are substantially affected by natural external forcing such as volcanic eruptions. Thus, the "SMTR estimate" method (the linear trend is removed in each ensemble member) is not appropriate for estimating their internal variability amplitudes (gray

and green plots in Figures 11a and 12a). This issue has already been mentioned in the other part. In contrast, the internal variability amplitude of the Niño-3.4 SST is close between the SMTR and MMMR estimates when using the 50 ensemble members (Figure 10a), despite that the natural external forcing such as volcanoes could potentially induce an interannual-scale forced response in the Niño-3.4 SST. This implies that such forced response is relatively small and thus the SMTR estimate could derive an accurate amplitude of the Niño-3.4 SST internal variability.

As for potential inaccuracies in the use of a linear trend such as noise influences, we think that these have been already implied as follows: "the second method, which requires a large ensemble, is more appropriate for determining the internal variability component since it is not contaminated by residuals from detrending methods."

l224: I got confused by the wording here. You refer to a "single-member estimate" but then proceed to discuss the method, which doesn't sound like a single member estimate at all. The description sounds like an "N member estimate". Suggest clarification.

We will change the wording to describe the two methods. After suggestion by reviewer #2, we will use "single-member-trend removed (SMTR) estimate" and "multi-member-mean removed (MMMR) estimate" through the manuscript.

l246: Presumably some measure has been chosen to quantify whether it has been "degraded". Suggest being clear by what measure you are using here.

We will remove the specific threshold number in the corresponding text. Instead, the text will be rewritten in a general way as follows: "a smaller ensemble size (e.g., N=10) results in a higher probability of the underestimated standard deviation by the MMMR estimate, demonstrating the necessity of a large ensemble to evaluate the internal variability amplitude."

l310: Again, it seems you need to have chosen some threshold to quantify whether the amplitude of the internal variability is underestimated. Suggest being clear about how you have determined that.

Similar to the above response, we will remove the specific number and rewrite the text in a general way as follows: "a large ensemble is necessary to avoid underestimating the amplitude of internal variability relative to the ensemble mean."

L315: There is an accompanying single forcing large ensemble for the CESM2-LE which I think would increase the number of years of simulation for CESM2 (https://www.cesm.ucar.edu/working-groups/climate/simulations/cesm2-single-forcing-le)

Thank you. We will denote the sum of the following simulations (32675 years):

- historical + SSP370: 100 members of 1850-2100 = 25100 years
- GHG only: 15 members of 1850-2050 = 1515 years
- Anthropogenic aerosol only: 20 members of 1850-2050 = 2020 years
- Biomass burning only: 15 members of 1850-2050 = 1515 years
- All but GHG, AAER and BMM: 15 members of 1850-2050 = 1515 years
- All but AAER: 10 members of 1850-2050 = 1010 years

Typo's/wording:

l58: suggest changing "and" between the reference to the biomass burning simulations and the greenhouse gas simulations to "or" since it is not both that are time evolving.

It will be changed.

l131: "TX" --> "Tx"

We will correct it.

l157: Here, and throughout, there's some inconsistency as to whether you refer to "ssp" or "SSP" and "ssp245" or "SSP-2.45".   Suggest being consistent.

In CMIP6, SSP2-4.5 indicates the concentration scenario, and ssp245 means experiments under the SSP2-4.5 scenario. For example, please see Gillett et al. (2016, https://doi.org/10.5194/gmd-

9-3685-2016). Although it may be slightly confusing, we will follow the usage of those terms in CMIP6.

l252: "variabilities than the best" --> "variabilities compared to the best"

We will correct it.

l319: "federation grid" --> "grid federation'

We will correct it.

---

## Author Comment (AC2)

Reply to reviewer #2

This paper is intended as a documentation paper for a new Large Ensemble of climate model simulations with MIROC6. This is an extensive set of new simulations, including historical & scenario simulations, as well as single forcing experiments. It presents the experimental design and overview of simulations, as well as presents some high-level analyses of global climate characteristics of this ensemble, in particular testing the linear additivity of the single-forcing experiments to make up the all-forcing simulations, and projected changes in ENSO variability.

The paper introduces a valuable new resource for the climate community, is well written, and presents some interesting first analyses of climate characteristics in this ensemble. This is a valuable contribution, and I would recommend publication subject to minor revisions. I have included my comments below. Comments 6) and 12) are slightly more substantial than the others but should be easy to address.

Thank you very much for your useful comments.

1) Do the 50 historical ensemble members of the LE include the CMIP6 historical ensemble members, or are they separate? Is the model version identical to CMIP6 (i.e., can both ensembles be merged)?

The LE includes the CMIP6 simulations, and the model version is identical to CMIP6. We will clarify it as "We increased the ensemble sizes of the CMIP6 simulations and performed an experiment recently proposed."

2) How were the initial conditions years selected / spaced out in the piControl run?

We will involve the list of initial conditions as a table.

3) Timeseries figures (1, 2, 6 & 7): it would be very helpful to draw a line at 0

We will draw a line at 0.

4) L105: It might be worth noting that the effect of hist-stratO3, hist-totalO3 and hist-lu are small but not zero especially in the 2nd half of the 20th century

We will rewrite this sentence as "Changes in stratospheric and tropospheric ozone (hist-stratO3 and hist-totalO3) and land-use-land-cover (hist-lu) have small effects on the global mean T and Tx: changes in stratospheric ozone and land-use land-cover have small cooling effects in the 2nd half of 20th century and changes in total ozone have small warming effects.".

5) L108: do you mean to say small trend or no trend? I would suggest either using "the P of the historical runs **only** has a small trend" or "has little trend" depending on the intended meaning

We will use "only has a small trend".

6) L117-126: The median of the blue bars (corresponding to the linear sum of single forcing experiments) is in some cases not near the middle of the orange bars, suggesting some non-linearities in the medians / ensemble means. There could still be important non-linearities here that would be evident in the ensemble means (or medians); overlap of the blue / orange ranges is not evidence of the absence of non-linearities in the ensemble means. This should be commented on here

Thank you for the useful advice. For the revised version, we have computed the max-min ranges of 1000 ensemble average values of randomly sampled ensemble members with replacement. The range of the ensemble mean of historical runs overlap with the range of the sum of the ensemble average of individual forcing experiments. Therefore we have not found any statistical evidence of non-linearities in the ensemble mean values.

7) L139 typo: faction -> fraction

We will correct it.

8) Figure 4: I would suggest using a title for each row e.g. deltaTx(hist)-deltaTx(hist-nat) to help the reader

We will add titles on Fig. 4.

9) L153-155: I don't think 'nearly disappear' for the aerosol signal by end of 21st century in T is quite correct: substantially reduced yes, nearly disappear I'm not sure – it looks like -0.2 degrees C by the end of the century. That's still about half of the strongest aerosol response in the late 20th century (8pprox.. -0.4C by eye)?

We will revise those lines as "Because aerosol emissions gradually decrease under SSP2-4.5 (Rao 2017, Lund 2019), the negative responses of T, Tx and Px significantly decrease. Although negative responses of P to aerosol forcing also decrease, apparent differences of P changes between ssp245 and ssp245-GHG remain until 2100 mainly due to the large sensitivity of P to aerosol forcing (Shiogama et al. 2010a, b)."

10) L 187: Internal variability (=variabilities) in plural sounds a bit unusual, I think you could use Internal variability singular here and in other places

We will use "Internal variability" in the revised manuscript.

11) L 210: single-ensemble – don't you mean single-member?

We will correct it.

12) The methods and terms used to describe the two methods are somewhat ambiguous (L190ff): I think it should be made clear that the single-member / multi-member name refers to the method for removing the forced response only, since the circle and crosses in Figures 10, 11 and 12 are both estimated from averages across multiple ensemble members. The authors could try using different names: multi-member-mean removed and single-member trend removed, for example

The authors also need to explicitly state how they calculate the variability once the forced response is removed in both methods: standard deviation across time for each member, and then

averaged across all ensemble members? Are you averaging standard deviations or variances?

L 198 is especially ambiguous: does the "best estimate" ensemble averages apply to both methods?

Thank you for the suggestion. We will change the wording to describe the two methods. After your suggestion, we will use "single-member-trend removed (SMTR) estimate" and "multi-member-mean removed (MMMR) estimate" through the manuscript.

The method for calculating the variability will be clarified as follows: "The standard deviation is derived across time for each member, and then averaged across ensemble members. We term the 50 member averages of the standard deviations derived by the MMMR estimate 'the best estimate' in this study."

13) L291: It might be worth stating explicitly that AGCMs cannot simulate changes in coupled modes of variability and SST patterns, and hence can only inform projections and attribution statements conditionally with respect to prescribed SST patterns

We will explain it as "while AGCM simulations can only inform projections and attribution statements conditionally with respect to prescribed SST patterns"

14) L297: perhaps worth inserting "future changes in
ENSO **variability"** or **"ENSO amplitude**"

We will correct it.

It is somewhat obvious, but the authors state in a number of places that larger ensembles are needed to detect changes between ssp119 and ssp126. This is certainly true, but it might be worth saying somewhere something to the effect of "unsurprisingly, the smaller the difference in forcing, the larger the ensemble needs to be to detect differences in the forced response. This is just a suggestion, the authors can choose to take it or leave it!

We will add "It is expected that larger ensembles are necessary to identify differences in climate

response for smaller differences of forcing." in the text.

---

## Author Response (AR1)

Reply to the editor,

Thank you very much. We have revised the manuscript to response to the comments from the reviewers. We have highlighted all the changes in the pink color in the manuscript, but I am sorry that I have not remained "MS word track-changes". We have checked the colors of our figures by using the Coblis. We hope the revised manuscript satisfies you and the reviewer.

Best regards,
Hideo Shiogama

Reply to the reviewer #1

This study introduces an impressive new dataset that is available for community use with the MIROC6 model.   This dataset consists of large ensembles of historical and future projections with multiple scenarios as well as single forcing simulations.   The primary purpose of the paper is to introduce these simulations but it also presents some cursory, but useful, analyses of changes in global temperature, precipitation and their extremes and some assessment of non-linearity in the single forcing simulations and analysis of the number of members required to detect changes or differences between scenarios.   Overall, I think this is a useful and well written study that introduces this important new dataset and I have only minor recommendations to consider before publication.

Thank you very much for your useful comments.

l83: It could be worth providing a bit more information about the piControl and the initialization dates.   Firstly, it could be worth stating whether the piControl is still drifting at this stage and some more specifics about the initialization dates e.g., were they spaced by a certain number of years?

The piControl run is stable (Fig. 3 of Tatebe et al. 2019). We involved the list of initial conditions in Table 2 (L83-85).

L87: It can sometimes be a bit confusing where biomass burning aerosols are represented.   I'm assuming that they are included in the anthropogenic aerosol contribution?   Even though there is a natural component to that.   It might be worth being clear about this.

Although biomass burning aerosol emissions include both anthropogenic and natural components, anomalies from piControl (involving the 1850 emission of biomass burning aerosols) can be used to estimate the anthropogenic component. (L91-92)

l120-126 and Fig 3: I'm not sure what the motivation is for doing this assessment of non-linearity by using only samplings of 1 member.   It may be that there is a true non-linearity but

you can only see it in the ensemble means.   You could do the same analysis but sample N members with replacement from each ensemble, where N is your original ensemble size, to determine whether there are any non-linearities that can be detected with the ensemble means.

Thank you for the useful advice. In this revised version, we computed the max-min ranges of 1000 ensemble average values of randomly sampled ensemble members with replacement. The range of the ensemble mean of historical runs overlap with the range of the sum of the ensemble average of individual forcing experiments. Therefore we did not find any statistical evidence of non-linearities in the ensemble mean values (L125-131).

l137: At the introduction to Fig 5, it might help readers to remind them what time period is being considered.   I think it's 2000-2020 minus 1850-1900?

We denoted "2000-2020 minus 1850-1900" in the caption of Fig. 5.

l147-151: I think this text is describing the behavior of the hist-nat+ssp245-nat run in Figure 6, but it's not entirely clear.   Maybe reference that part of the figure when referring to the solar and volcanic contributions.

This text is describing the solar and volcanic forcing in the future simulations (2015-). We made it clear in L153 as "For future simulations from 2015, volcanic …" and drew only the period of 2015-2100 in Fig. 6.

l210: It seems like another possibility beyond the interannual external forcings is inacuracies in the use of a linear trend?   If so, that could be mentioned too.

Thank you for the question. First, we would like to clarify the sentence (L217), "… the forced response of the Niño-3.4 SST is not sensitive to interannual external forcing, such as volcanoes". As seen in Figures 1 and 2, the ensemble averages (forced responses) of the global mean annual temperature and precipitation are substantially affected by natural external forcing such as volcanic eruptions. Thus, the "SMTR estimate" method (the linear trend is removed in each ensemble member) is not appropriate for estimating their internal variability amplitudes

(gray and green plots in Figures 11a and 12a). This issue has already been mentioned in L259-265. In contrast, the internal variability amplitude of the Niño-3.4 SST is close between the SMTR and MMMR estimates when using the 50 ensemble members (Figure 10a), despite that the natural external forcing such as volcanoes could potentially induce an interannual-scale forced response in the Niño-3.4 SST. This implies that such forced response is relatively small and thus the SMTR estimate could derive an accurate amplitude of the Niño-3.4 SST internal variability.

As for potential inaccuracies in the use of a linear trend such as noise influences, we think that these have been already implied in L203-204 as follows: "the second method, which requires a large ensemble, is more appropriate for determining the internal variability component since it is not contaminated by residuals from detrending methods."

l224: I got confused by the wording here. You refer to a "single-member estimate" but then proceed to discuss the method, which doesn't sound like a single member estimate at all. The description sounds like an "N member estimate". Suggest clarification.

We have changed the wording to describe the two methods. After suggestion by reviewer #2, we now use "single-member-trend removed (SMTR) estimate" and "multi-member-mean removed (MMMR) estimate" through the manuscript (e.g., L198-200).

l246: Presumably some measure has been chosen to quantify whether it has been "degraded". Suggest being clear by what measure you are using here.

We removed the specific threshold number in the corresponding text. Instead, the text has been rewritten in a general way as follows: "a smaller ensemble size (e.g., N=10) results in a higher probability of the underestimated standard deviation by the MMMR estimate, demonstrating the necessity of a large ensemble to evaluate the internal variability amplitude." (L253-255)

l310: Again, it seems you need to have chosen some threshold to quantify whether the amplitude of the internal variability is underestimated. Suggest being clear about how you have determined that.

Similar to the above response, we have removed the specific number and rewritten the text in a general way as follows: "a large ensemble is necessary to avoid underestimating the amplitude of internal variability relative to the ensemble mean." (L318-319)

L315: There is an accompanying single forcing large ensemble for the CESM2-LE which I think would increase the number of years of simulation for CESM2 (https://www.cesm.ucar.edu/working-groups/climate/simulations/cesm2-single-forcing-le)

Thank you. We computed the sum of the following simulations (32675 years) in L324:

- historical + SSP370: 100 members of 1850-2100 = 25100 years
- GHG only: 15 members of 1850-2050 = 1515 years
- Anthropogenic aerosol only: 20 members of 1850-2050 = 2020 years
- Biomass burning only: 15 members of 1850-2050 = 1515 years
- All but GHG, AAER and BMM: 15 members of 1850-2050 = 1515 years
- All but AAER: 10 members of 1850-2050 = 1010 years

Typo's/wording:

l58: suggest changing "and" between the reference to the biomass burning simulations and the greenhouse gas simulations to "or" since it is not both that are time evolving.

Changed (L58).

l131: "TX" --> "Tx"

Corrected (L136).

l157: Here, and throughout, there's some inconsistency as to whether you refer to "ssp" or "SSP" and "ssp245" or "SSP-2.45".   Suggest being consistent.

In CMIP6, SSP2-4.5 indicates the concentration scenario, and ssp245 means experiments under the SSP2-4.5 scenario. For example, please see Gillett et al. (2016, https://doi.org/10.5194/gmd-9-3685-2016). Although it may be slightly confusing, we followed the usage of those terms in CMIP6.

l252: "variabilities than the best" --> "variabilities compared to the best"

Corrected (L260).

l319: "federation grid" --> "grid federation'

Corrected (L328).

Reply to reviewer #2

This paper is intended as a documentation paper for a new Large Ensemble of climate model simulations with MIROC6. This is an extensive set of new simulations, including historical & scenario simulations, as well as single forcing experiments. It presents the experimental design and overview of simulations, as well as presents some high-level analyses of global climate characteristics of this ensemble, in particular testing the linear additivity of the single-forcing experiments to make up the all-forcing simulations, and projected changes in ENSO variability.

The paper introduces a valuable new resource for the climate community, is well written, and presents some interesting first analyses of climate characteristics in this ensemble. This is a valuable contribution, and I would recommend publication subject to minor revisions. I have included my comments below. Comments 6) and 12) are slightly more substantial than the others but should be easy to address.

Thank you very much for your useful comments.

1) Do the 50 historical ensemble members of the LE include the CMIP6 historical ensemble members, or are they separate? Is the model version identical to CMIP6 (i.e., can both ensembles be merged)?

The LE includes the CMIP6 simulations, and the model version is identical to CMIP6. We clarified it in L81-82: "We increased the ensemble sizes of the CMIP6 simulations and performed an experiment recently proposed."

2) How were the initial conditions years selected / spaced out in the piControl run?

We involved the list of initial conditions in Table 2 (L83-85).

3) Timeseries figures (1, 2, 6 & 7): it would be very helpful to draw a line at 0

We drew a line at 0.

4) L105: It might be worth noting that the effect of hist-stratO3, hist-totalO3 and hist-lu are small but not zero especially in the 2nd half of the 20th century

We rewrote this sentence as "Changes in stratospheric and tropospheric ozone (hist-stratO3 and hist-totalO3) and land-use-land-cover (hist-lu) have small effects on the global mean T and Tx: changes in stratospheric ozone and land-use land-cover have small cooling effects in the 2nd half of 20th century and changes in total ozone have small warming effects." (L108-109).

5) L108: do you mean to say small trend or no trend? I would suggest either using "the P of the historical runs **only** has a small trend" or "has little trend" depending on the intended meaning

We used "only has a small trend" (L112).

6) L117-126: The median of the blue bars (corresponding to the linear sum of single forcing experiments) is in some cases not near the middle of the orange bars, suggesting some non-linearities in the medians / ensemble means. There could still be important non-linearities here that would be evident in the ensemble means (or medians); overlap of the blue / orange ranges is not evidence of the absence of non-linearities in the ensemble means. This should be commented on here

Thank you for the useful advice. In this revised version, we computed the max-min ranges of 1000 ensemble average values of randomly sampled ensemble members with replacement. The range of the ensemble mean of historical runs overlap with the range of the sum of the ensemble average values of the individual forcing experiments. Therefore we did not find any statistical evidence of non-linearities in the ensemble mean values (L125-131).

7) L139 typo: faction -> fraction

Corrected (L144).

8) Figure 4: I would suggest using a title for each row e.g. deltaTx(hist)-deltaTx(hist-nat) to help the reader

We added titles on Fig. 4.

9) L153-155: I don't think 'nearly disappear' for the aerosol signal by end of 21st century in T is quite correct: substantially reduced yes, nearly disappear I'm not sure – it looks like -0.2 degrees C by the end of the century. That's still about half of the strongest aerosol response in the late 20th century (9pprox.. -0.4C by eye)?

We revised those lines as "Because aerosol emissions gradually decrease under SSP2-4.5 (Rao 2017, Lund 2019), the negative responses of T, Tx and Px significantly decrease. Although negative responses of P to aerosol forcing also decrease, apparent differences of P changes between ssp245 and ssp245-GHG remain until 2100 mainly due to the large sensitivity of P to aerosol forcing (Shiogama et al. 2010a, b)." (L158-161)

10) L 187: Internal variability (=variabilities) in plural sounds a bit unusual, I think you could use Internal variability singular here and in other places

We used "Internal variability" in this revised manuscript.

11) L 210: single-ensemble – don't you mean single-member?

Corrected (L218).

12) The methods and terms used to describe the two methods are somewhat ambiguous (L190ff): I think it should be made clear that the single-member / multi-member name refers to the method for removing the forced response only, since the circle and crosses in Figures 10, 11 and 12 are both estimated from averages across multiple ensemble members. The authors could try using different names: multi-member-mean removed and single-member trend removed, for example

The authors also need to explicitly state how they calculate the variability once the forced response is removed in both methods: standard deviation across time for each member, and then

averaged across all ensemble members? Are you averaging standard deviations or variances?

L 198 is especially ambiguous: does the "best estimate" ensemble averages apply to both methods?

Thank you for the suggestion. We have changed the wording to describe the two methods. After your suggestion, we now use "single-member-trend removed (SMTR) estimate" and "multi-member-mean removed (MMMR) estimate" through the manuscript (e.g., L198-200).

The method for calculating the variability has been clarified as follows: "The standard deviation is derived across time for each member, and then averaged across ensemble members. We term the 50 member averages of the standard deviations derived by the MMMR estimate 'the best estimate' in this study." (L204-206)

13) L291: It might be worth stating explicitly that AGCMs cannot simulate changes in coupled modes of variability and SST patterns, and hence can only inform projections and attribution statements conditionally with respect to prescribed SST patterns

We explained it in L300-301: "while AGCM simulations can only inform projections and attribution statements conditionally with respect to prescribed SST patterns"

14) L297: perhaps worth inserting "future changes in
ENSO **variability"** or **"ENSO amplitude**"

Corrected (L306).

It is somewhat obvious, but the authors state in a number of places that larger ensembles are needed to detect changes between ssp119 and ssp126. This is certainly true, but it might be worth saying somewhere something to the effect of "unsurprisingly, the smaller the difference in forcing, the larger the ensemble needs to be to detect differences in the forced response. This is just a suggestion, the authors can choose to take it or leave it!

We added "It is expected that larger ensembles are necessary to identify differences in climate

response for smaller differences of forcing." in L182-183.